# Physiological and Transcriptome Analyses of Photosynthesis in Three Mulberry Cultivars within Two Propagation Methods (Cutting and Grafting) under Waterlogging Stress

**DOI:** 10.3390/plants12112066

**Published:** 2023-05-23

**Authors:** Yong Li, Jin Huang, Cui Yu, Rongli Mo, Zhixian Zhu, Zhaoxia Dong, Xingming Hu, Chuxiong Zhuang, Wen Deng

**Affiliations:** 1Cash Crops Research Institute, Hubei Academy of Agricultural Sciences, Wuhan 430064, China; 2College of Life Sciences, South China Agricultural University, Guangzhou 510642, China

**Keywords:** mulberry, waterlogging, photosynthesis, gene regulation

## Abstract

Mulberry is a valuable woody plant with significant economic importance. It can be propagated through two main methods: cutting and grafting. Waterlogging can have a major impact on mulberry growth and can significantly reduce production. In this study, we examined gene expression patterns and photosynthetic responses in three waterlogged mulberry cultivars propagated through cutting and grafting. Compared to the control group, waterlogging treatments reduced levels of chlorophyll, soluble protein, soluble sugars, proline, and malondialdehyde (MDA). Additionally, the treatments significantly decreased the activities of ascorbate peroxidase (APX), peroxidase (POD), and catalase (CAT) in all three cultivars, except for superoxide dismutase (SOD). Waterlogging treatments also affected the rate of photosynthesis (Pn), stomatal conductance (Gs), and transpiration rate (Tr) in all three cultivars. However, no significant difference in physiological response was observed between the cutting and grafting groups. Gene expression patterns in the mulberry changed dramatically after waterlogging stress and varied between the two propagation methods. A total of 10,394 genes showed significant changes in expression levels, with the number of differentially expressed genes (DEGs) varying between comparison groups. GO and KEGG analysis revealed important DEGs, including photosynthesis-related genes that were significantly downregulated after waterlogging treatment. Notably, these genes were upregulated at day 10 in the cutting group compared to the grafting group. In particular, genes involved in carbon fixation were significantly upregulated in the cutting group. Finally, cutting propagation methods displayed better recovery capacity from waterlogging stress than grafting. This study provides valuable information for improving mulberry genetics in breeding programs.

## 1. Introduction

Waterlogging stress is a major hindrance to plant growth and can result in significant yield losses in many plants [1,2,3]. Waterlogging creates hypoxic conditions due to the slow diffusion of molecular oxygen in water, leading to various morphological and cellular acclimation responses [4,5]. Studies on hypoxia in crops have shown that it causes rapid changes in gene expression and cellular metabolism [6,7]. While hypoxia affects energy metabolism and root zone hypoxia is a key component of waterlogging stress, field observations have also shown reduced growth rates, photosynthesis rates, and stomatal conductivity in waterlogged plants [8]. It has been established that waterlogging stress is more complex than mere altered energy metabolism in plants. In response to waterlogging stress, plants exhibit both short-term and long-term adaptations. In the short-term, plants alter their physiological processes by reducing stomatal conductance and net photosynthetic rate [9]. In the long-term, plants regulate the expression of hundreds of relevant genes, consequently modifying individual phenotypes and some morphological and anatomical features [10,11,12]. One adaptation strategy is to change structural arrangement of stem cells and cell organelles [13,14]. The hypertrophy of lenticels and formation of aerenchyma facilitate plant oxygenation and contribute to maintaining root aerobic respiration and water uptake under waterlogging conditions [15]. Waterlogging-adapted species are characterized by higher density of xylem vessels, larger size of vessels, and a greater number of secondary meristem cells compared with waterlogging-sensitive species [14].

As photosynthesis is affected by waterlogging stress, which in turn affects plant production, it is crucial to understand the physiological and genetic regulation of waterlogged plants [16,17,18,19]. In plants, genetically regulated hormones such as ethylene, abscisic acid, and gibberellic acid play important roles in responding to environmental stress [20,21]. The transcriptional and physiological interplay in response to waterlogging stress may provide key insights into plant waterlogging tolerance [22,23]. However, the level and degree of tolerance may vary by species or abiotic pressures and requires further study. Investigating regulatory pathways and interactions in crops such as wheat, rice, soybeans, and mulberry will provide valuable resources for enhancing economically and agriculturally important traits in waterlogged areas.

Mulberry (*Morus* spp.) is a perennial woody plant with significant economic importance. Its nutritious fruits are widely used in pharmaceutical and traditional Chinese medicine [24,25]. Notably, over 60% of the total cost of cocoon production is spent on mulberry production alone [26]. The primary goal of improving mulberry germplasm is to develop new cultivars with high leaf yield, fruit quality, pest resistance, and tolerance to various abiotic stresses such as drought, waterlogging, and salt. In agricultural practice, grafting and cutting are two important propagation methods for mulberry [27,28]. However, little information is available on the waterlogging tolerance of mulberry propagated through these methods. While physiological responses at the cellular level have been studied in many crops [24,29], the physiological and genetic regulation of photosynthesis in response to waterlogging stress in mulberry remains unclear, particularly across different cultivars. In order to investigate the differences in waterlogging tolerance and response mechanisms among different mulberry cultivars and between different propagation methods (grafting and cutting) under waterlogging stress, and to provide scientific recommendations for mulberry breeding, we conducted this study. In this study, we examined the photosynthetic responses to waterlogging stress in three widely grown mulberry fruit cultivars (AY, SG, and ZZ) from the Yangtze River basin and used RNA-seq to analyze gene expression patterns under waterlogging stress. We also compared the waterlogging stress tolerance of mulberry propagated through cutting and grafting methods to identify the ideal propagation method for waterlogging tolerance. This study provides valuable information for improving mulberry varieties in waterlogged areas.

## 2. Results

### 2.1. Comparisons of Osmotic Regulatory Substances between Cutting and Grafting Groups in Three Cultivars

After waterlogging treatments, the levels of chlorophyll, soluble protein, soluble sugars, proline, and MDA in the waterlogged groups were significantly lower than those in the control groups in all three mulberry cultivars (Figure 1). Overall, levels of soluble protein, soluble sugar, proline, and MDA gradually decreased over 20 days. Notably, waterlogging stress reduced the levels of these osmotic regulatory substances. Levels of osmotically-regulating substances also differed significantly between the cutting and grafting groups in the ZZ and SG cultivars. In the AY cultivar, significant differences were observed between the cutting and grafting groups for MDA, proline, and chlorophyll levels, suggesting that different propagation methods may affect waterlogging stress tolerance through regulation of osmotic substances. Similar trends were observed in the other two cultivars.

### 2.2. Dynamics of Enzyme Activities after Waterlogging Treatments

We found that waterlogging treatments significantly reduced APX, POD, and CAT activities (Figure 2). Interestingly, SOD activities in the waterlogged treatment groups were significantly higher than those in the control groups in all three cultivars. SOD activities also gradually increased within 20 days after waterlogging treatments. No significant difference in enzyme activities was observed between the cutting and grafting groups in any of the three mulberry cultivars. A smaller difference between propagation methods was observed for the AY cultivar compared to the other two cultivars (SG and ZZ).

### 2.3. Dynamic Changes of Photosynthetic Characters in Three Mulberry Cultivars after Waterlogging Treatment

The rates of photosynthesis (Pn) for the three mulberry cultivars from 6:00 a.m. to 6:00 p.m. are shown in Figure 3. In all three cultivars, the Pn values for the waterlogged treatment groups were significantly lower than those for the control groups, indicating that waterlogging stress can impair photosynthesis in mulberry (Figure 3A). In the control groups for the AY cultivar, the grafted groups showed significantly higher Pn values than the cutting groups, suggesting that the cutting method may cause physiological damage to this cultivar (Figure 3A). Exposure to waterlogging resulted in a decrease in Pn, and no apparent peak was observed. No difference was observed between the cutting and grafting methods in the waterlogged treatment groups. The dynamics of stomatal conductance (Gs) showed that waterlogging treatment affected stomatal conductivity in all three mulberry cultivars (Figure 3B). In the AY cultivar, the grafted groups showed higher Gs values than the cutting groups, but a completely different trend was observed for the ZZ cultivar (Figure 3B). Interestingly, waterlogging stress appeared to decrease intercellular CO2 concentration (Ci) in the early stages, but no significant difference was observed after 10:00 a.m. (Figure 3C). The transpiration rate (Tr) curves for all three cultivars showed that waterlogging produced a strong effect on leaf transpiration (Figure 3D). The grafting and cutting methods also affected the transpiration rate in the AY and SG cultivars under waterlogging stress, with grafted groups showing higher Tr values in these two cultivars.

To further understand the dynamics of photosynthetic properties in the three mulberry cultivars after waterlogging, we measured the indices of Pn, Gs, Ci, and Tr over 20 days following waterlogging treatments. The curves for these indices are shown in Appendix A. Overall, Pn gradually increased as the mulberry plants recovered from waterlogging stress, although there was a slight decrease during the first three days. Waterlogging treatments lowered Pn in all three cultivars, and no significant difference was observed between the grafting and cutting groups (Appendix A). Similar results were obtained for Gs and Tr (Appendix A). Interestingly, the Ci curves showed that waterlogging stress produced only a minor effect on Ci in mulberry and on the propagation methods (Appendix A).

### 2.4. The Number of DEGs in Cut and Grafted Mulberry Responding to Waterlogging Stress

To understand the gene expression patterns and molecular basis of waterlogging tolerance in mulberry propagated through cutting and grafting, we performed RNA-seq on 36 samples from both propagation methods at 0-day recovery (D0), 3-day recovery (D3), and 10-day recovery (D10) after waterlogging. After quality control of the raw reads, a total of 233.37 Gb of clean reads were mapped to the reference genome with mapping ratios ranging from 71.37% to 76.99%. The FPKM values of the genes were calculated, and the pairwise Pearson correlation coefficients between the three biological replicates were above 0.9 (Appendix A). Principal component analysis (PCA) clearly showed clustering patterns for the three biological replicates (Appendix A), indicating high variability between biological replicates. All samples were divided into two groups according to control and waterlogging (Appendix A).

To identify differentially expressed genes (DEGs) between waterlogged and control groups and between cutting and grafting groups, we performed nine pairwise comparisons. The total number of up- and down-regulated genes varied between comparisons (Appendix A). For both cutting and grafting groups, the number of DEGs was over 4000 at D0, then quickly decreased at D3 before increasing again at D10. The number of DEGs between cutting and grafting groups at D0, D3, and D10 under waterlogging treatment was small and did not change significantly (Figure 4 and Appendix A). At D0, D3, and D10, 202 common DEGs were identified between grafted and cut mulberry after waterlogging stress (Figure 4A), while in the control group, 741 and 280 common DEGs were identified after waterlogging in grafted (Figure 4B) and cut (Figure 4C) mulberry, respectively. The largest number of unique DEGs from cut and grafted mulberry was observed at D0 with 1103 and 594 DEGs, respectively (Figure 4D). To understand the dynamic changes in mulberry, we performed another eight pairwise comparisons and found that the number of DEGs varied between D0 vs. D3 and D3 vs. D10 under the same conditions (Appendix A). A total of 108 and 83 common DEGs were identified in cut and grafted mulberry under waterlogging and control conditions, respectively (Appendix A). Under waterlogging stress, we identified 164 and 76 DEGs in cut and grafted mulberry, respectively, while under control conditions only 42 and 31 DEGs were identified in cut and grafted mulberry, respectively (Appendix A). In total, we identified 10,394 common DEGs in all 17 pairwise comparisons. These results indicate that waterlogging stress produces a major impact on mulberry growth and that cutting, grafting, and waterlogging all continuously affect the genetic processes in mulberry. Furthermore, cutting and grafting can enhance the waterlogging tolerance of mulberry.

### 2.5. Dynamics of Photosynthesis-Related Gene Expression in Mulberry with Waterlogging Stress

We performed GO enrichment analysis on the DEGs and found that several GO terms, including thylakoid, photosynthesis, chloroplast, photosystem, photosynthetic membrane, and chloroplast thylakoid, were significantly enriched in the pairwise comparisons (Appendix A). Several key metabolic pathways, including photosynthesis, carbon metabolism, sesquiterpenoid and triterpenoid biosynthesis, carbon fixation in photosynthetic organisms, and porphyrin and chlorophyll metabolism were also significantly enriched for the DEGs (Appendix A).

We selected DEGs identified between the waterlogged treatment and control groups to clarify mulberry’s genetic response to waterlogging stress. Most DEGs were related to energy metabolism, carbohydrate metabolism, and lipid metabolism (Figure 5A). Most importantly, waterlogging stress directly or indirectly caused the downregulation of key genes related to photosynthesis, such as photosystem II genes (*PsbA*, *PsbB*, *PsbC*, etc.), photosystem I genes (*PsaA*, *PsaB*, *PsaD*, etc.), cytochrome b6/f complex genes (*PetB*, *PetD*, *PetA*, *PetC*, etc.), and photosynthetic electron transport genes (*PetE*, *PetF*, *PetJ*). These DEGs were detected at time points D0, D3, and D10 (Figure 5B). The gene expression patterns showed that most photosynthesis-related genes were downregulated after waterlogging treatment, particularly at D0 (Figure 5C). As the plants recovered from waterlogging stress at D3 and D10, some photosynthesis-related genes were expressed at levels similar to those in the control groups (Figure 5D,E).

### 2.6. Propagation Methods Affect Photosynthesis-Related Gene Expression Responding to Waterlogging Stress

To focus on differences in gene regulation between grafted and cut groups, we selected DEGs identified between these groups. According to KEGG pathway enrichment analysis, several key pathways were identified at both D3 and D10 time points, including flavonoid biosynthesis, circadian rhythm, and cutin, suberin, and wax biosynthesis. For example, chalcone synthase genes (*CHS1*, *CHS2*, *CHS3*), *HD3A*, and *HD3B* showed higher expression levels in the cutting groups. Additionally, genes enriched in the MAPK pathway (*MPK3*, *MPK14*, *MKK9*, *CML45*) were upregulated in the cutting groups at the D3 time point (Figure 6A). Interestingly, the photosynthetic pathway was enriched with DEGs at the D10 time point but not at the D3 time point (Figure 6B). All photosynthesis-related genes were found to be upregulated in the cutting groups compared to the grafted groups, suggesting that the cutting groups had better photosynthetic recovery ability after waterlogging treatments. Similarly, genes associated with carbon fixation were also upregulated in the cutting groups compared to the grafted groups.

### 2.7. Validation of Photosynthesis-Related Genes Expression with qPCR Method

We selected ten photosynthesis-related genes from the DEGs for qPCR analysis. Compared to the control, our result showed that all genes were down-regulated after the waterlogging stress except for *FEDA*, which was consistent with the RNA-seq data (Figure 7). *Lhca* and *Lhcb* belong to light-harvesting chlorophyll a/b binding (LHC) superfamily, which plays critical roles in photosynthesis [30]. The expression change of *Lhcb* gene family normally influenced plant growth and development under abiotic stress [31]. The down-regulation of *AtLhcb4, AtLhcb5,* and *AtLhcb6* expression leads to the accumulation of higher levels of superoxide and more severe oxidative stress, which further disrupts photoprotection [32,33]. Overexpression of the tomato *Lhcb2* gene in tobacco alleviated photo-oxidation of PSII and enhanced tobacco tolerance to chilling stress [34]. We hypothesize that down-regulation of these genes also may affect the photo-oxidation in photosynthesis to alter waterlogging tolerance in mulberry. *FEDA* was a ferredoxin and reduced expression of the *FEDA* gene in *Arabidopsis* leaves destroyed by redox-regulated adaptations in the photosynthetic system [35]. *FEDA* was extremely up-regulated after waterlogging recovery at D3 in our study (Figure 7). Therefore, we hypothesize that the *FEDA* gene may regulate the oxidative adaptation of the mulberry photosystem by increasing its expression in response to waterlogging stress.

## 3. Discussion

Waterlogging is an important water-related stress that can damage plants by rapidly reducing the rate of photosynthesis and stomatal conductivity [36]. Photosynthesis is highly sensitive to water stress, and limitations in photosynthetic carbon metabolism have been analyzed in crops [37,38,39]. Photosynthetic cells are highly sensitive to oxidative stress and have robust antioxidant systems to protect plants from oxidative damage [36]. At the genetic level, the negative effects of waterlogging stress on leaves may result from oxidative damage to important molecules due to an imbalance between the production of activated oxygen and its metabolism in plants [40,41]. Previous studies have documented the effects of water-related stress on various crops, including mulberry [28,42,43]. However, the mechanisms by which waterlogging stress inhibits photosynthesis in mulberry have not been extensively investigated. A better understanding of the mechanisms that allow plants to adapt to waterlogging stress and maintain growth and productivity during periods of waterlogging will ultimately aid in the selection of waterlogging-tolerant cultivars. Efficient approaches to identifying waterlogging-resistant genotypes and understanding the key periods during which plants can tolerate waterlogging has been a key goal for plant researchers.

### 3.1. Dynamic Changes in Physiological Indices of Mulberry under Waterlogging Stress

Under waterlogging conditions, plants produce antioxidants, flavonoids, and secondary metabolites that play a role in protecting the plant by detoxifying reactive oxygen species (ROS) and stabilizing proteins and amino acids. These compounds help the plant to cope with abnormal conditions caused by waterlogging [44]. Superoxide dismutase (SOD), peroxidase (POD), catalase (CAT), and ascorbate peroxidase (APX) are key enzymes involved in the detoxification of reactive oxygen species (ROS) in plants. These enzymes work together to remove toxic oxygen species and protect the plant from oxidative damage under stress conditions. SOD catalyzes the conversion of superoxide radicals into hydrogen peroxide and oxygen, while CAT, POD, and APX work to remove hydrogen peroxide by converting it into water and oxygen. This coordinated action helps to maintain the redox balance within the cell and prevent oxidative damage to cellular components such as lipids, proteins, and nucleic acids. The activity of these enzymes can be modulated by various factors, including changes in environmental conditions, developmental stages, and the presence of other stressors. Understanding the regulation of these enzymes and their role in plant stress responses can provide valuable insights into the mechanisms underlying plant adaptation to changing environments.

In this study, we found that after waterlogging treatment in mulberry, the activities of APX, CAT, and POD decreased, except for SOD. Additionally, chlorophyll content was also reduced by waterlogging (Figure 2). Chlorophyll plays a crucial role in capturing light energy and converting it into chemical energy through photosynthesis. As such, a reduction in chlorophyll content can directly lead to a decrease in the rate of photosynthesis in plants [45]. Furthermore, waterlogging resulted in significant decreases in leaf malondialdehyde, soluble protein, soluble sugars, and proline production in mulberry. Additionally, there was a reduction in leaf photosynthetic rate, stomata conductance, and transpiration rate. These results are largely consistent with previous studies [46,47] that reported a reduction in leaf protective enzymes and photosynthetic rate when plants were subjected to waterlogging stress. Notably, we found that unlike other protective enzymes, SOD activities gradually increased after waterlogging treatment. This finding suggests that SOD plays a crucial role in responding to active oxygen activities caused by waterlogging in mulberry. It is possible that the increase in SOD activity stimulates the cellular protective mechanism to mitigate damage. Meanwhile, we found that the early reduction in photosynthetic efficiency after waterlogging treatment in mulberry could last for 7 days and recover after that time.

The observed decrease of the content of these enzymes may indicate an alteration in the plant’s ability to cope with oxidative stress caused by waterlogging. Previous studies have shown that waterlogging can lead to an increase in ROS production, due to changes in cellular metabolism and energy production [48]. Under normal conditions, plants have a range of antioxidant defense mechanisms to cope with ROS, including the enzymes APX, POD, and CAT [49]. However, under stress conditions such as waterlogging, these defense mechanisms may be overwhelmed, leading to oxidative damage and changes in enzyme activity [50]. In addition to changes in ROS production and antioxidant defense mechanisms, waterlogging can also affect other aspects of plant physiology. For example, waterlogging can lead to changes in nutrient uptake and transport, as well as alterations in hormone signaling and gene expression [51]. These changes can affect plant growth and development, as well as their ability to cope with stress.

In conclusion, our results suggest that waterlogging can alter the antioxidant defense mechanisms of mulberry trees, as indicated by the decrease in the content of APX, POD, and CAT. Further studies are needed to elucidate the mechanisms underlying this response and to determine its impact on plant growth and productivity.

### 3.2. Cutting Propagation Methods Displayed Better Recovery Capacity from Waterlogging Stress than Grafting

Grafting and cutting are widely used methods of plant propagation and play a crucial role in improving the yield and quality of crop trees and vegetables. Grafting can typically alter several physiological and biochemical reactions between the rootstock and scion, affecting the growth, development, and resilience of plants [52,53]. Previous studies have reported that grafting improves photosynthetic capacity [54] and antioxidant system activity of crops under salt stress [55].

However, in this study, we found that the mulberry grafting method showed no advantage in reducing the inhibitory effect of waterlogging stress on photosynthesis compared to the cutting method (Figure 3). Waterlogging stress occurs when the soil becomes saturated with water, depriving the roots of oxygen and making it difficult for the plant to take up nutrients. This condition can inhibit photosynthesis and reduce plant growth. It seems that in this case, the cutting method was more effective in helping mulberry trees recover from waterlogging stress.

At the genetic level, photosynthesis in the early phase after waterlogging treatment showed no difference between grafting and cutting groups. However, after 10 days of waterlogging treatment, photosynthesis-related genes were up-regulated in the cutting groups (Figure 4). Photosynthesis, carbon metabolism, sesquiterpenoid and triterpenoid biosynthesis, carbon fixation in photosynthetic organisms, and porphyrin and chlorophyll metabolism were significantly enriched for the DEGs under waterlogging stress (Appendix A).

Waterlogging stress directly or indirectly caused the downregulation of key genes related to photosynthesis, such as photosystem II genes (*PsbA*, *PsbB*, *PsbC*, etc.), photosystem I genes (*PsaA*, *PsaB*, *PsaD*, etc.), cytochrome b6/f complex genes (*PetB*, *PetD*, *PetA*, *PetC*, etc.), and photosynthetic electron transport genes (*PetE*, *PetF*, *PetJ*) (Figure 5).

Interestingly, the cutting groups showed higher expression levels of chalcone synthase genes (*CHS1*, *CHS2*, *CHS3*), *HD3A*, *HD3B* and the MAPK pathway (*MPK3*, *MPK14*, *MKK9*, *CML45*) than the grafting group (Figure 6). Chalcone synthase (CHS) is a crucial rate-limiting enzyme in the flavonoid biosynthetic pathway that plays an important role in regulating plant growth, development, and abiotic stress tolerance [56]. The MAPK pathway is also involved in stress responses in plants. It is a signaling pathway that can be activated by various stimuli, including abiotic stresses such as drought, cold, and salt stress. Activation of the MAPK pathway can lead to changes in gene expression that help the plant cope with the stress. Abiotic stress can cause damage to plants at the cellular level and disrupt their normal growth and development. In response to stress, plants activate various defense mechanisms to protect themselves. One such mechanism is the production of secondary metabolites such as flavonoids, which can help protect the plant against stress-induced damage [57,58]. The increased expression of chalcone synthase genes under stress may be part of this defense mechanism. The *HD3A* and *HD3B* genes are involved in the regulation of flowering time in plants [59]. Abiotic stress can affect the timing of flowering and disrupt the normal reproductive cycle of plants. The increased expression of *HD3A* and *HD3B* genes under stress may be part of the plant’s response to ensure proper flowering and reproduction despite the stressful conditions.

In addition to the fact that the expression of some key genes in the cutting group was higher than the grafting group, we also found that cutting groups had better recovery capacity from waterlogging stress than grafted mulberry. It should be noted that the genetic trait of different rootstocks leading to varying waterlogging tolerance may result in different recovery abilities of grafted mulberry trees in response to waterlogging or other environmental stresses, and the more combinations should be investigated in further studies. 

## 4. Materials and Methods

### 4.1. Plants and Sample Preparation

The experiment was conducted at the Industrial Crops Institute of the Hubei Academy of Agricultural Sciences. In this study, three representative mulberry fruit cultivars (AY, SG, and ZZ) were selected for propagation by grafting and cutting. Grafting and cutting trials were carried out in March 2019 using two-year-old healthy mulberry plants. After grafting and cutting, the mulberry seedlings were planted in freshly prepared pots (28 × 20 × 20 cm) containing well-dried, pulverized garden soil, decomposed sand, and well-rotted manure in a ratio of 5:3:2. The seedlings were cared for consistently. The experiment was carried out with 10 replicates per variety.

Waterlogging treatments were carried out in June 2019. Before the treatments, each pot was placed in a 150 cm diameter trough. Waterlogging treatments were performed by filling the outer tank with water up to 5 cm above the sand surface. The control groups were watered normally. The waterlogging treatment was carried out with 10 repetitions. Mulberry samples were collected in the morning after 0, 3, 7, 12, and 18 days.

### 4.2. Measurements of Osmotic Regulation Substances and Chlorophyll Content

To measure the dynamics of osmotically regulating substances in mulberry, we analyzed the levels of soluble protein, sugar, proline, and malondialdehyde (MDA) in mulberry leaves. 

The soluble protein content was measured using the Coomassie brilliant blue method. The amount of 0.1 mL of sample extract was taken into a test tube and 5 mL of G250 reagent was added. After mixing well for 2 min, distilled water was used as the blank, and absorbance was measured at 595 nm. The absorbance value was recorded, and the protein concentration was obtained through the standard curve. The soluble protein content (ng/g) was further calculated.

The soluble sugar content was measured using the anthrone colorimetric method. The amount of 0.5 mL of extract was taken into a 10 mL test tube, with distilled water as the control. Subsequently, 0.5 mL of distilled water and 5 mL of anthrone reagent were added and mixed well. Absorbance was then measured at a wavelength of 620 nm. The optical density value was recorded, and the corresponding sugar content (μg/g) was obtained through the standard curve.

Proline content was measured using the sulfosalicylic acid method. Next, 2 mL of supernatant was taken, and 2 mL of ice acetic acid plus 3 mL of coloring solution were added. The mixture was boiled in a water bath for 40 min and cooled. The amount of 5 mL of toluene was then added to the solution, which was shaken well. After standing for layering, the toluene layer of absorbance was measured at 520 nm, with toluene as the zero adjustment to calculate the proline content (ng/g).

Malondialdehyde (MDA) content was measured using the thiobarbituric acid method. The absorbance of the supernatant extract at 450 nm, 532 nm, and 600 nm was measured. The MDA content in the tissue (nmol/g) was then calculated.

Chlorophyll content was measured using the acetone extraction method. Fresh mulberry leaves in the amount of 0.5 g were weighed and placed in a mortar. Some quartz sand and 80% acetone were added and ground into a homogenate. The homogenate was filtered into a volumetric flask. The mortar, glass rod, funnel, and filter paper were rinsed with 80% acetone until all the chlorophyll was rinsed into the volumetric flask and finally diluted to 10 mL. The mixture was mixed well, and the absorbance was measured at 645 nm and 663 nm to calculate the chlorophyll content (ng/g).

### 4.3. Measurements of Enzyme Activities

Fresh leaves with three replicates were collected from the upper shoots to measure the activities of four enzymes (SOD, CAT, POD, APX).

Superoxide dismutase (SOD) activity was measured using the nitroblue tetrazolium (NBT) method. The amount of 0.1 mL of crude enzyme extract was taken for enzyme activity measurement. Phosphate buffer solution replaced enzyme solution as the control, with one light control and one dark control. Each reaction solution was added to a 10 mL test tube to a final volume of 3.3 mL. The test tube and control test tube were placed under a 4000 lx daylight lamp at a temperature of 25–35 °C for 20–30 min. The dark control reacted for the same length of time in the dark. The reaction was immediately terminated by covering the test tube after it was over. During measurement, the solution was diluted appropriately, and the dark control was used as the blank. The absorbance value was measured with a UV-visible spectrophotometer at a wavelength of 550 nm and recorded to calculate SOD enzyme activity (U/g).

Catalase (CAT) activity was measured using the UV absorption method. Phosphate buffer in the amount of 0.1 mL of 2% H_2_O_2_ and 2 mL was taken into a 1 cm quartz cuvette, and 0.1 mL of crude enzyme extract was added. The mixture was mixed well at room temperature and immediately measured for changes in optical density within 5 min using a UV-visible spectrophotometer at a wavelength of 240 nm until the optical density reduction per minute reached stability. CAT activity (U/g) was then calculated.

Peroxidase (POD) activity was measured using the guaiacol method [60]. Enzyme activity was measured with 0.1 mL of crude enzyme extract. According to the reaction system, 2.9 mL of 0.05 mol/L phosphate buffer solution, 0.5 mL of 2% H_2_O_2_, 0.1 mL of 2% guaiacol solution, and 0.1 mL of enzyme solution were added respectively. The mixture was mixed well at room temperature, and absorbance was measured at 420 nm until the optical density reduction per minute reached stability. POD activity (U/g) was then calculated.

Ascorbic acid peroxidase (APX) activity was measured using an Elisa kit. The absorbance was measured at a wavelength of 290 nm using an enzyme-linked immunosorbent assay instrument. The plant ascorbic acid peroxidase activity (U/g) in the sample was calculated through the standard curve.

### 4.4. Measurements of Photosynthetic Characters 

The net photosynthesis rate (Pn), stroma conductance (Gs), intercellular CO_2_ concentration (Ci), and transpiration rate (Tr) of mulberry leaves were measured simultaneously from 6:00 a.m. to 6:00 p.m. using a LI-6400XT portable photosynthetic analyzer manufactured by LI-COR (Lincoln, NE, USA). The measurements were carried out with three repetitions. The diurnal fluctuations in photosynthesis were measured from 12 May to 13 May. Leaves were selected from three well-lit top shoots of mulberry trees. A leaf with normal function was selected from each shoot. Measurements were taken once every 2 h and repeated three times each time.

### 4.5. Statistical Analysis

Means and standard deviation of replicates were calculated using Microsoft (Redmond, WA, USA) Office Excel 2010. Statistical analysis was performed using SPSS19.0 software (SPSS, Inc., Chicago, IL, USA) with *p*-value < 0.05 as significant difference in this study. 

### 4.6. RNA-Seq of Mulberry Leaves

After waterlogging treatment, mulberry leaves from two groups (cutting and grafting) were collected at 0, 3, and 10 days, with corresponding controls set for each time point. Total RNA was extracted using RNAiso (TaKaRa, Dalian, China). Three biological replicates were performed for RNA-seq. The concentration and quality were checked with NanoDrop 2000 (Life Technology, Waltham, MA, USA) and Agilent 2100 Bioanalyzer System (Agilent, Santa Clara, CA, USA). After quality control of the RNA samples, a total of 36 libraries were constructed using the NEB library kit, according to the instructions. Sequencing was performed using the BGI MGIseq 2000 system. The raw reads generated by the MGIseq platform were filtered using fastp [61] with default parameters.

The HISAT2 was used to align the clean data to the mulberry reference genome [62], then RSEM software was used to calculate gene expression levels based the alignment file. Based on Fragments Per Kilobase of exon model per million mapped fragments (FPKM) files, the R function cor and prcomp were used to perform Pairwise Pearson correlation coefficient and principal component analysis (PCA), respectively. Gene expression patterns were evaluated using the TCseq package (https://github.com/MengjunWu/TCseq, accessed on 10 December 2022). The differentially expressed genes (DEGs) were identified using the DEseq2 program [63] with the criterion of |Fold Change| > 2 and FDR < 0.05. The Venn Diagram and UpSetR R package were used to draw the Venn diagram for these DEGs. Gene ontology (GO), and Kyoto Encyclopedia of Genes and Genomes (KEGG) pathway enrichment analysis of DEGs were performed using OmicShare tools (https://www.omicshare.com, accessed on 10 December 2022).

### 4.7. Validation of RNA-Seq Data

First strand cDNA was synthesized using PrimeScript RT reagent kit (TaKaRa, China). RT-qPCR was performed with Light Cycler 480 (Roche, Basel, Switzerland) in 20 L with SYBR Premix Ex Taq Kit (TakaRa, China). Reaction conditions were 95 for 3 min followed by 40 cycles of 94 for 10 s, 55 for 10 s, and 72 for 30 s. Gene expression levels were calculated relative to the expression levels of -actin and GADPH using the 2-Ct method. Primers were designed using the NCBI primer design program. The sequences of the primers are shown in Appendix A.

## 5. Conclusions

In conclusion, our study revealed that the osmotically regulating substances and enzyme activities in three cultivars showed little difference between the cutting and grafting groups, with the exception of the AY cultivar which showed a clear difference from the ZZ and SG cultivars. Our results also demonstrated that waterlogging can affect the photosynthetic rate in mulberry, with a gradual increase in photosynthetic rate observed as mulberry recovered from waterlogging stress over a 20-day period. At the genetic level, waterlogging stress was found to directly or indirectly cause the down-regulation of key genes associated with photosynthesis, such as *petC* and *petE*. Interestingly, cutting groups were found to have a better ability to recover from waterlogging stress than grafted mulberries. These findings provide valuable insights into the underlying mechanisms of dual-method mulberry propagation in responding to waterlogging stress and highlight the potential for developing waterlogging-tolerant mulberry cultivars.

Our study contributes to the understanding of the physiological and genetic responses of mulberry to waterlogging stress and provides a foundation for future research on improving waterlogging tolerance in this economically important crop. The striking finding that cutting groups can better recover from waterlogging stress than grafted mulberries offers a promising avenue for further investigation and may have important implications for mulberry cultivation in waterlogged areas.

## Figures and Tables

**Figure 1 plants-12-02066-f001:**
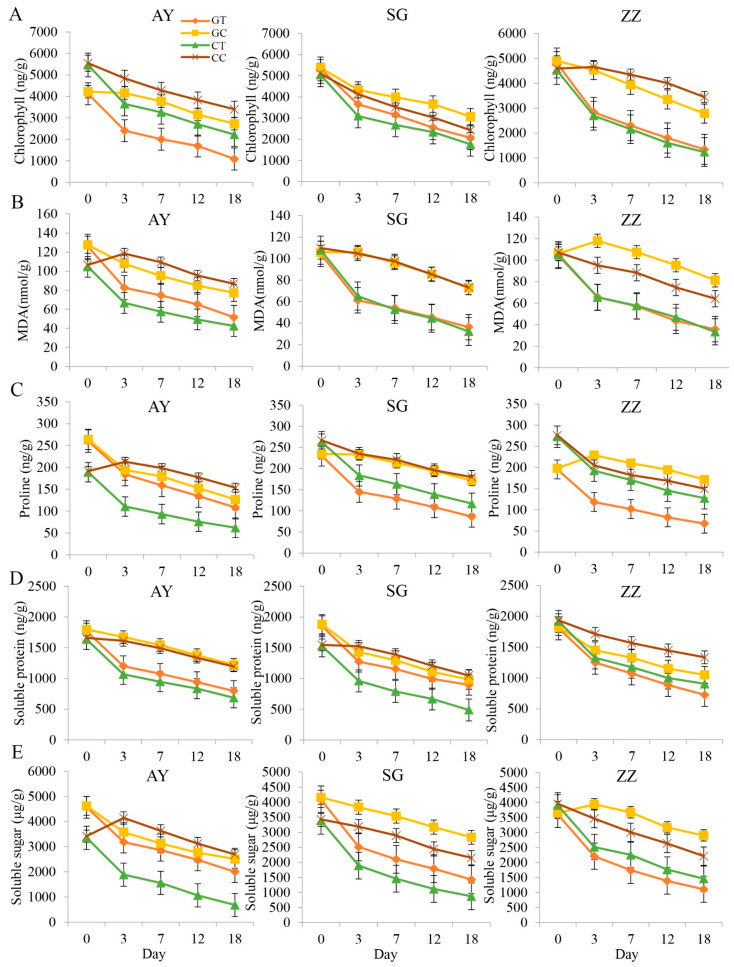
The contents of osmotic regulation substances in three cultivars between waterlogging and control groups, including chlorophyll (**A**), MDA (**B**), proline (**C**), soluble protein (**D**), and soluble sugar (**E**) in three mulberry cultivars (AY, SG and ZZ) between waterlogging and control groups. CC, CT, GC, and GT in the legend indicate cut mulberry under control and waterlogging treatment, and grafted mulberry under control and waterlogging treatment, respectively.

**Figure 2 plants-12-02066-f002:**
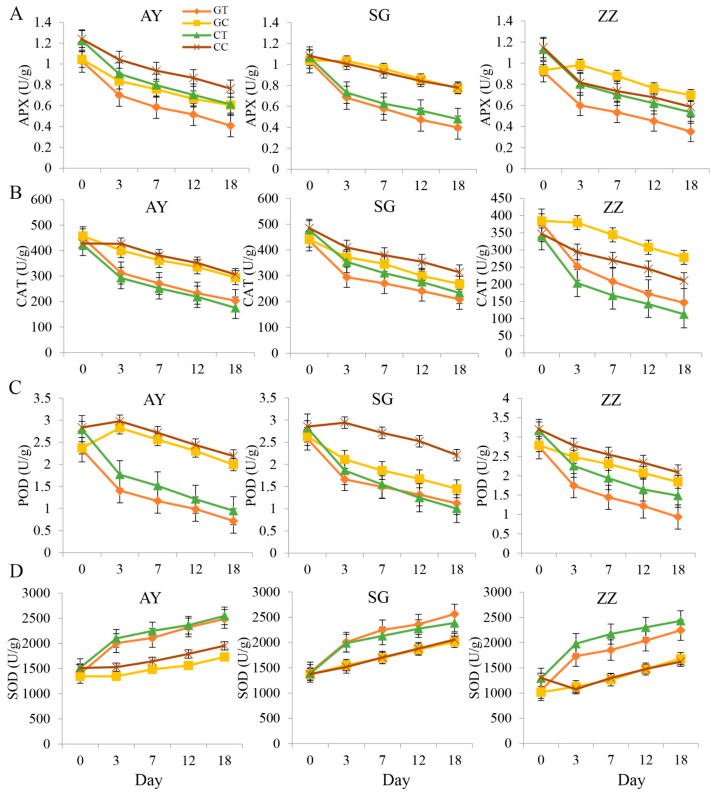
The changes of enzyme activities in three cultivars after waterlogging treatments, including APX (**A**), CAT (**B**), POD (**C**), and SOD (**D**) in three mulberry cultivars (AY, SG and ZZ) between waterlogging and control groups. CC, CT, GC, and GT in the legend indicate cut mulberry under control and waterlogging treatment, and grafted mulberry under control and waterlogging treatment, respectively.

**Figure 3 plants-12-02066-f003:**
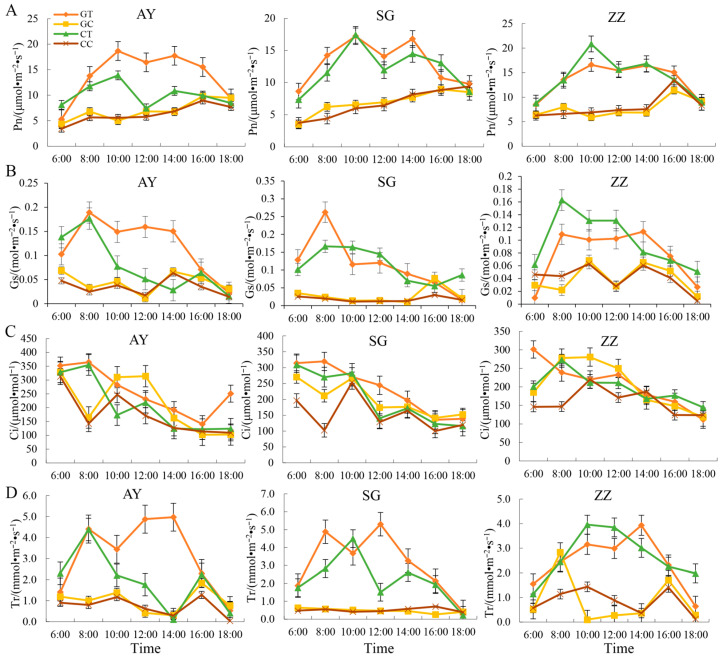
The dynamics of photosynthetic indexes, including Pn (**A**), Gs (**B**), Ci (**C**), and Tr (**D**) in three mulberry cultivars (AY, SG and ZZ) between waterlogging and control groups. CC, CT, GC, and GT in the legend indicate cut mulberry under control and waterlogging treatment, and grafted mulberry under control and waterlogging treatment, respectively.

**Figure 4 plants-12-02066-f004:**
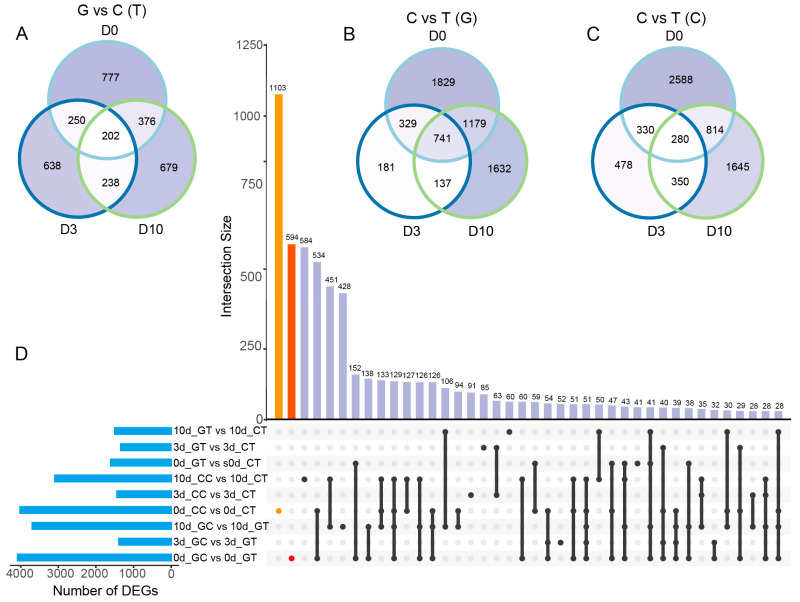
Venn diagram and UpSet indicated the number of differentially expressed genes (DEGs) per contrast: divided into three groups including grafting and cutting under waterlogging treatments (**A**); control and waterlogging treatments in grafting (**B**) and cutting (**C**); upset summary for nine pairwise comparisons (**D**). Numbers in intersections represent the number of DEGs shared in the intersection contrasts.

**Figure 5 plants-12-02066-f005:**
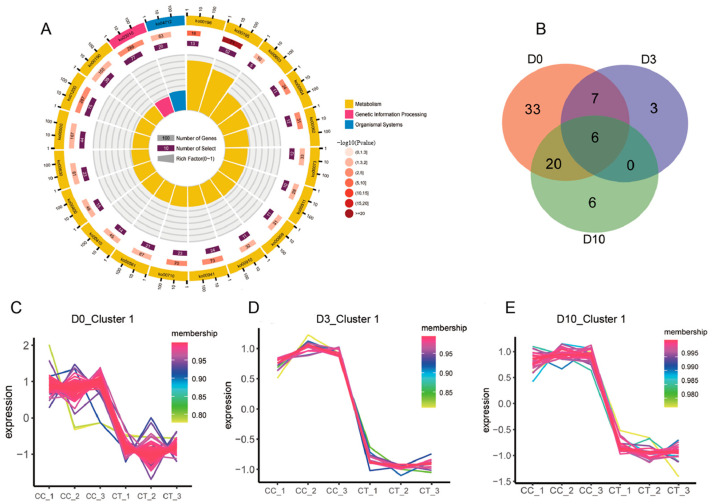
The KEGG enrichment analysis for DEGs between waterlogging and control group (**A**); Venn diagram of the photosynthesis-related DEGs from three comparisons (**B**); gene expression trend of the photosynthesis-related genes in the waterlogging treatments and control group in D0 (**C**), D3 (**D**), and D10 (**E**), respectively.

**Figure 6 plants-12-02066-f006:**
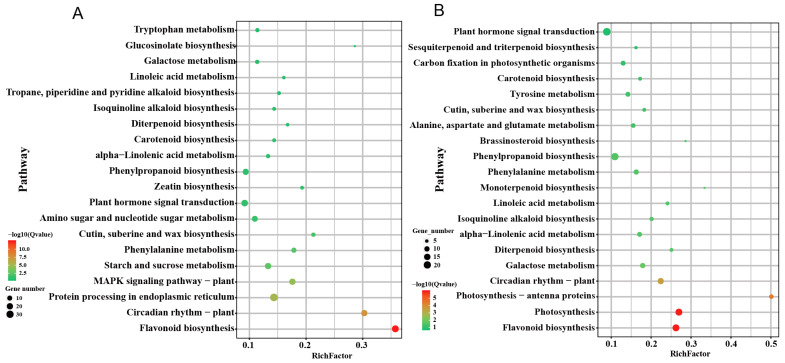
Top 20 enriched KEGG pathways between grafted and cutting groups under waterlogging stress at day 3 (**A**) and day 10 (**B**). The number of differentially expressed genes (DEGs) in each pathway is counted, and the enrichment factors and *p*-values are displayed.

**Figure 7 plants-12-02066-f007:**
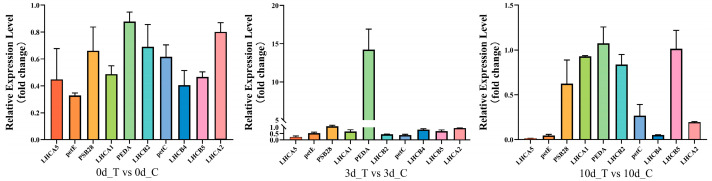
Validation of RNA-seq data for the expression of 10 photosynthesis-related genes using the qPCR method in cutting-propagated mulberry cultivars under waterlogging and control conditions at 0, 3, and 10 days.

## Data Availability

The details of the DEGs (Appendix A) have been deposited in figshare (https://doi.org/10.6084/m9.figshare.14343932.v1).

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
