# Peer review of "Physiological and Transcriptome Analyses of Photosynthesis in Three Mulberry Cultivars within Two Propagation Methods (Cutting and Grafting) under Waterlogging Stress"

_plants, 2023, doi:10.3390/plants12112066_

Round 1
Reviewer 1 Report (Previous Reviewer 1)
Review 2
The manuscript of Li et al “Physiological and transcriptome analyses of photosynthesis in three mulberry cultivars within two propagation methods (cutting and grafting) under waterlogging stress“ has undergone significant revision, which undoubtedly contributed to its improvement.
However, despite language fixes, it still contains some obvious errors. This is especially relevant for part 2.7 (lines. (241-258) In particular,
Line 240 please, change “all gene” for “all genes”
Line 247 please, delete “a” before “critical roles”
Line 248 please, change “influence” for “influenced”
Lines 249-250 “The down-regulated expression of AtLhcb4, AtLhcb5 and AtLhcb6 disrupted oxidative stress and photoprotection” Probably only the photoprotection is meant under conditions of oxidative stress
Line 251 please, change “alleviate” for “alleviated”
Line 251 please, change “down-regulated” for “down-regulation”
Lines 368-369 please, change “In addition to the expression of some key genes in the cutting group is higher than the grafting group” for “In addition to the fact that the expression of some key genes in the cutting group was higher than the grafting group
Lines 398,403,410 – “colorimetric” should be replaced by “absorbance was measured”
Line 457 please, change “stoma” for “stroma”
Lines 517-518 “This study provided some clues to the underlying mechanisms of dual-method mulberry in responding to waterlogging stress. - Probably dual-method of mulberry propagation is meant
I would also advise to indicate the duration of waterlogging treatments in“Materials and methods”
Text editing is desirable
Author Response
Please see the attachment.

Reviewer 2 Report (New Reviewer)
General comments
I have read the manuscript: Entitle: Physiological and transcriptome analyses of photosynthesis in three mulberry cultivars within two propagation methods (cutting and grafting) under waterlogging stress written by Yong Li et. al., for publication of Plants MDPI. In this study, the author examined gene expression patterns and photosynthetic responses of mulberry cultivars under waterlogging conditions. Result showed waterlogging treatments reduced levels of chlorophyll, soluble protein, soluble sugars, proline, and malondialdehyde (MDA), and significantly decreased ascorbate peroxidase (APX), peroxidase (POD), and catalase (CAT). Waterlogging treatments also affected the rate of photosynthesis (Pn), stomatal conductance (Gs), and transpiration rate (Tr) in all three cultivars. Gene expression patterns in mulberry changed dramatically after waterlogging stress. The overall research is well conducted.
This study provides valuable information for 34 improving mulberry genetics in breeding programs. In this sense this manuscript is much valuable. However, author should significantly improve this manuscript for journal acceptance. I found a lack of story connection and some lack of potential references (some I suggested below). Overall after I evaluate and request the author for this manuscript as a “MAJOR REVISION”. If author well address the comments and improve the manuscript this article may accept for the publication.
Major Suggestions
1) Introduction: Author should be use more potentially references in this section. Waterlogging stress reducing the rate of photosynthesis and stomatal conductivity. This article http://doi.org/10.1016/j.envexpbot.2020.104111 well deals with the morphological and physiological and anatomical traits under the long-term flooding conditions which are very useful to deal with your discussion section.
2) Hypothesis of the study: The author well presented the research aim or objective of the study clearly in Ln. 73-77, but research hypothesis is not clearly mentioned and is not connected with the objectives. The research hypothesis should be very clear and connected each other (with objectives) because without appropriate literature, questions, or hypotheses the entire introduction section will not be clear.
3) Discussion: Author should Improve subtitle 3.1 Author should further deal the antioxidant and secondary metabolites formation especially SOD, POD CAT, and other secondary metabolites under waterlogging condition by referring this reference https://doi.org/10.1016/j.agrformet.2022.109071 “Under waterlogging plant produce antioxidant, flavonoids, and secondary metabolites play to the role for protecting the plant for detoxifying ROS and protect the abnormal condition (i.e. waterlogging) and protein and amino acid stabilization”.
Some other comments
4) Line 40: Author should improve the introduction further by including the short- and long-term effect of waterlogging and how plant show defense mechanism/symptoms from both of these stress, should refer the previously published waterlogging related articles.
5) Line 239 and 260: Author should improve the legend of each figure. In scientific writing induvial figure should speck all the presented component all via. Figure itself or by the legend. Please rephrase the figure 6 and Fig 7 legend and check others figures as well for the consistency.
6) Line no. 378: Author should include information related to chlorophyll and how it performs for the physiological traits (photosynthesis). 3.1 sub-title is quite good to include related information. Refer the article DOI:10.1016/j.scienta.2018.11.021 “physiological performance especially higher the photosynthesis is due to increase of Chlorophyll content because Chl. help to capture the better light and higher amount of light due to Chl. then higher possibility of Pn because of conversation of light energy change into the chemical energy”.
7) Line 503 (Conclusion): Author should be mentioned the independent conclusion section instead of the summary. Further should not be repetitive in the abstract or a summary of the results section. I would love to read striking points and take-home messages that will linger in the readers’ minds. What is the novelty, how does the study elucidate some questions in this field, and the contributions the paper may offer to the scientific community?
8) Line 529 (References): please double-check the citations, their style, spell check, and other grammatical errors. moreover, the author should cut the old and less matching literature and include the latest literature some of them are above.
Good Luck!
Round 2
Reviewer 2 Report (New Reviewer)
Dear Author
I have read the revised manuscript plants2368455. Entitled: Physiological and transcriptome analyses of photosynthesis in three mulberry cultivars within two propagation methods (cutting and grafting) under waterlogging stress for publication in Plants. This is the second submission made by the author. The author addressed all the questions and suggestions that I raised the issue in the review of the original manuscript. I satisfy the author’s revisions. Author improves their hypothesis and well connected with the research objectives in this time. This manuscript improved the flow of writing, which was comparatively shallow in the original version but in this revised copy author very well addressed all the quarries and suggestions. Before accepting this manuscript if there is anything needed to be revised by the author, especially English grammar, or spell check, I request this manuscript is currently in “Minor Revision” and the author may correct any further grammatical errors (if any) the author may improve in this stage.
Thank you.
I request this manuscript is currently in “Minor Revision” and the author may correct any further grammatical errors (if any) the author may improve in this stage.
Author Response
Please see the attachment.

This manuscript is a resubmission of an earlier submission. The following is a list of the peer review reports and author responses from that submission.
Round 1
Reviewer 1 Report
Review
The manuscript of Y. Li and colleagues describes the effect of waterlogging stress and its consequences on transcriptome and physiological parameters in three mulberry cultivars under two propagation methods, cutting and grafting. In general, the idea of the manuscript is interesting, the authors present a lot of data of undoubted interest for the genetic improvement of mulberry. Notably, they show that waterlogging stress can affect photosynthesis and key metabolic pathways associated with it. They also demonstrate that cuttings have better recovery capacity than grafted mulberry from waterlogging stress unlike other environmental stresses.
However, the data obtained deserve a more thorough discussion and comparison with the results of exposure to other types of stress. In particular, it would be good to get an explanation why the content of proline and malondialdehyde and a number of enzymes decrease during flooding, while they usually increase under oxidative stress. At the same time the increase in SOD activity implies elevation in ROS production. Perhaps the determination of the content of superoxide radicals and hydrogen peroxide, accompanied by data from transcriptomic analysis on genes encoding protective enzymes, could contribute to a better understanding of the role the aforementioned enzymes in waterlogging stress.
I would also advise to place figure 5S in the main part, justifying the choice of genes for validating the results of the RNA-sec analysis and explaining in more detail how they are related to the mechanisms of inhibition of photosynthesis.
The results obtained would be more convincing if the authors presented, at least in a hypothetical form, considerations regarding the reasons for better recovery after stress of cuttings and the possible impact of combinations of rootstalk-scion genotypes.
I have also several minor comments on the text of the manuscript before it could be accepted for publication
Minor issues:
Line 22: please, change expect to except
Lines 140-143 Fig 3 The legend is given twice, please remove the extra figure caption
Line 397 please, list the genes to finish the sentence
Reviewer 2 Report
This manuscript entitled "Physiological and transcriptome analyses of photosynthesis in three mulberry cultivars within two propagation methods (cutting and grafting) under waterlogging stress"; could be good for publication in Plants (ISSN 2223-7747). This may be interesting, but some important points need to be resolved. Importantly, a study must provide a critical analysis of the data. In other words, you must assess whether specific data published really stand up to scientific scrutiny. In order to achieve the above, you must clearly define your specific aims and objectives. So in your study you must develop a critical appraisal of the state of the art. This is an essential element of any article. There are important scientific questions (both conceptual and methodological) which need to be addressed with the primary studies. A study must highlight this. The introduction, which is written in clear language, covers a number of relevant issues. Information are noteworthy, and not are correct supported by similar results from the specialty (see WOS:000229981900029, WOS:000244133900015, WOS:000232584300013). Try to rewrite the abstract and conclusions, I also recommend the nuance of the introduction, the way of working is not very well explained, the procedure is tedious and unsustainable. For this reason, I recommend that the authors try to use more sustainable methodologies, the interpretation of the results can be improved/ reformulated.
Reviewer 3 Report
The work presented by Yu et al., seeks to gain some understanding of the physiological and transcriptomic dynamics of three mulberry cultivars that propagated in two ways in response to waterlogging. However, the quality of the work is questionable, which is unsuitable to be published. First, the writing is hard to be followed. The work should be redrafted extensively. Second, the English language should be improved largely. Third, the data need to be better presented.
Some examples of above-mentioned problems are listed
1. The 1st sentence of the Abstract is hard to be understood. Many abbreviations were used in the Abstract without presenting the full description at the 1st appearance.
2. Line 43-44, check grammer;
3. Line 50-52, the statement is weak
4. Figure1 and 2 do not show the units of the Y-axis. How many bio replicates were used? Technique replicates?
Round 2
Reviewer 2 Report
I don't think that the authors took into account the recommendations of the reviewers